# Polynomial time algorithms for dual volume sampling

**Chengtao Li**
MIT
ctli@mit.edu

**Stefanie Jegelka**
MIT
stefje@csail.mit.edu

**Suvrit Sra**
MIT
suvrit@mit.edu

## Abstract

We study *dual volume sampling*, a method for selecting $k$ columns from an $n \times m$ short and wide matrix ($n \leq k \leq m$) such that the probability of selection is proportional to the *volume* spanned by the rows of the induced submatrix. This method was proposed by Avron and Boutsidis (2013), who showed it to be a promising method for column subset selection and its multiple applications. However, its wider adoption has been hampered by the lack of polynomial time sampling algorithms. We remove this hindrance by developing an exact (randomized) polynomial time sampling algorithm as well as its derandomization. Thereafter, we study dual volume sampling via the theory of real stable polynomials and prove that its distribution satisfies the "Strong Rayleigh" property. This result has numerous consequences, including a provably fast-mixing Markov chain sampler that makes dual volume sampling much more attractive to practitioners. This sampler is closely related to classical algorithms for popular experimental design methods that are to date lacking theoretical analysis but are known to empirically work well.

## 1 Introduction

A variety of applications share the core task of selecting a subset of columns from a short, wide matrix $A$ with $n$ rows and $m > n$ columns. The criteria for selecting these columns typically aim at preserving information about the span of $A$ while generating a well-conditioned submatrix. Classical and recent examples include experimental design, where we select observations or experiments [38]; preconditioning for solving linear systems and constructing low-stretch spanning trees (here $A$ is a version of the node-edge incidence matrix and we select edges in a graph) [6, 4]; matrix approximation [11, 13, 24]; feature selection in $k$-means clustering [10, 12]; sensor selection [25] and graph signal processing [14, 41].

In this work, we study a randomized approach that holds promise for all of these applications. This approach relies on sampling columns of $A$ according to a probability distribution defined over its submatrices: the probability of selecting a set $S$ of $k$ columns from $A$, with $n \leq k \leq m$, is

$$P(S; A) \propto \det(A_S A_S^\top), \tag{1.1}$$

where $A_S$ is the submatrix consisting of the selected columns. This distribution is reminiscent of *volume sampling*, where $k < n$ columns are selected with probability proportional to the determinant $\det(A_S^\top A_S)$ of a $k \times k$ matrix, i.e., the squared volume of the parallelepiped spanned by the selected columns. (Volume sampling does *not* apply to $k > n$ as the involved determinants vanish.) In contrast, $P(S; A)$ uses the determinant of an $n \times n$ matrix and uses the volume spanned by the *rows* formed by the selected columns. Hence we refer to $P(S; A)$-sampling as *dual volume sampling (DVS)*.

**Contributions.** Despite the ostensible similarity between volume sampling and DVS, and despite the many practical implications of DVS outlined below, efficient algorithms for DVS are not known and were raised as open questions in [6]. In this work, we make two key contributions:

– We develop polynomial-time randomized sampling algorithms and their derandomization for DVS. Surprisingly, our proofs require only elementary (but involved) matrix manipulations.

- We establish that $P(S; A)$ is a *Strongly Rayleigh* measure [8], a remarkable property that captures a specific form of negative dependence. Our proof relies on the theory of real stable polynomials, and the ensuing result implies a provably fast-mixing, practical MCMC sampler. Moreover, this result implies concentration properties for dual volume sampling.

In parallel with our work, [16] also proposed a polynomial time sampling algorithm that works efficiently in practice. Our work goes on to further uncover the hitherto unknown "Strong Rayleigh" property of DVS, which has important consequences, including those noted above.

## 1.1 Connections and implications.

The selection of $k \geq n$ columns from a short and wide matrix has many applications. Our algorithms for DVS hence have several implications and connections; we note a few below.

**Experimental design.** The theory of optimal experiment design explores several criteria for selecting the set of columns (experiments) $S$. Popular choices are

$$S \in \operatorname{argmin}_{S \subseteq \{1,\ldots,m\}} J(A_S), \text{ with } \quad J(A_S) = \|A_S^\dagger\|_F = \|(A_S A_S^\top)^{-1}\|_F \text{ (A-optimal design) },$$

$$J(A_S) = \|A_S^\dagger\|_2 \text{ (E-optimal design) }, \quad J(A_S) = -\log \det(A_S A_S^\top) \text{ (D-optimal design).} \tag{1.2}$$

Here, $A^\dagger$ denotes the Moore-Penrose pseudoinverse of $A$, and the minimization ranges over all $S$ such that $A_S$ has full row rank $n$. A-optimal design, for instance, is statistically optimal for linear regression [38].

Finding an optimal solution for these design problems is NP-hard; and most discrete algorithms use local search [33]. Avron and Boutsidis [6, Theorem 3.1] show that dual volume sampling yields an approximation guarantee for both A- and E-optimal design: if $S$ is sampled from $P(S; A)$, then

$$\mathbb{E}\left[\|A_S^\dagger\|_F^2\right] \leq \frac{m-n+1}{k-n+1}\|A^\dagger\|_F^2; \quad \mathbb{E}\left[\|A_S^\dagger\|_2^2\right] \leq \left(1 + \frac{n(m-k)}{k-n+1}\right)\|A^\dagger\|_2^2. \tag{1.3}$$

Avron and Boutsidis [6] provide a polynomial time sampling algorithm only for the case $k = n$. Our algorithms achieve the bound (1.3) in expectation, and the derandomization in Section 2.3 achieves the bound deterministically. Wang et al. [43] recently (in parallel) achieved approximation bounds for A-optimality via a different algorithm combining convex relaxation and a greedy method. Other methods include leverage score sampling [30] and predictive length sampling [45].

**Low-stretch spanning trees and applications.** Objectives 1.2 also arise in the construction of low-stretch spanning trees, which have important applications in graph sparsification, preconditioning and solving symmetric diagonally dominant (SDD) linear systems [40], among others [18]. In the node-edge incidence matrix $\Pi \in \mathbb{R}^{n \times m}$ of an undirected graph $G$ with $n$ nodes and $m$ edges, the column corresponding to edge $(u, v)$ is $\sqrt{w(u,v)}(e_u - e_v)$. Let $\Pi = U\Sigma Y$ be the SVD of $\Pi$ with $Y \in \mathbb{R}^{n-1 \times m}$. The stretch of a spanning tree $T$ in $G$ is then given by $St_T(G) = \|Y_T^{-1}\|_F^2$ [6]. In those applications, we hence search for a set of edges with low stretch.

**Network controllability.** The problem of sampling $k \geq n$ columns in a matrix also arises in network controllability. For example, Zhao et al. [44] consider selecting control nodes $S$ (under certain constraints) over time in complex networks to control a linear time-invariant network. After transforming the problem into a column subset selection problem from a short and wide controllability matrix, the objective becomes essentially an E-optimal design problem, for which the authors use greedy heuristics.

**Notation.** From a matrix $A \in \mathbb{R}^{n \times m}$ with $m \gg n$ columns, we sample a set $S \subseteq [m]$ of $k$ columns ($n \leq k \leq m$), where $[m] := \{1, 2, \ldots, m\}$. We denote the singular values of $A$ by $\{\sigma_i(A)\}_{i=1}^n$, in decreasing order. We will assume $A$ has full row rank $r(A) = n$, so $\sigma_n(A) > 0$. We also assume that $r(A_S) = r(A) = n$ for every $S \subseteq [m]$ where $|S| \geq n$. By $e_k(A)$, we denote the $k$-th elementary symmetric polynomial of $A$, i.e., the $k$-th coefficient of the characteristic polynomial $\det(\lambda I - A) = \sum_{j=0}^N (-1)^j e_j(A) \lambda^{N-j}$.

## 2 Polynomial-time Dual Volume Sampling

We describe in this section our method to sample from the distribution $P(S; A)$. Our first method relies on the key insight that, as we show, the marginal probabilities for DVS can be computed in polynomial time. To demonstrate this, we begin with the partition function and then derive marginals.

## 2.1 Marginals

The partition function has a conveniently simple closed form, which follows from the Cauchy-Binet formula and was also derived in [6].

**Lemma 1** (Partition Function [6]). *For $A \in \mathbb{R}^{n \times m}$ with $r(A) = n$ and $n \leq |S| = k \leq m$, we have*

$$Z_A := \sum_{|S|=k, S \subseteq [m]} \det(A_S A_S^\top) = \binom{m-n}{k-n} \det(AA^\top).$$

Next, we will need the marginal probability $P(T \subseteq S; A) = \sum_{S:T \subseteq S} P(S; A)$ that a given set $T \subseteq [m]$ is a subset of the random set $S$. In the following theorem, the set $T_c = [m] \setminus T$ denotes the (set) complement of $T$, and $Q^\perp$ denotes the orthogonal complement of $Q$.

**Theorem 2** (Marginals). *Let $T \subseteq [m]$, $|T| \leq k$, and $\varepsilon > 0$. Let $A_T = Q\Sigma V^\top$ be the singular value decomposition of $A_T$ where $Q \in \mathbb{R}^{n \times r(A_T)}$, and $Q^\perp \in \mathbb{R}^{n \times (n-r(A_T))}$. Further define the matrices*

$$B = (Q^\perp)^\top A_{T_c} \in \mathbb{R}^{(n-r(A_T)) \times (m-|T|)},$$

$$C = \begin{bmatrix} \frac{1}{\sqrt{\sigma_1^2(A_T)+\varepsilon}} & 0 & \cdots \\ 0 & \frac{1}{\sqrt{\sigma_2^2(A_T)+\varepsilon}} & \cdots \\ \vdots & \vdots & \ddots \end{bmatrix} Q^\top A_{T_c} \in \mathbb{R}^{r(A_T) \times (m-|T|)}.$$

*Let $Q_B \mathrm{diag}(\sigma_i^2(B))Q_B^\top$ be the eigenvalue decomposition of $B^\top B$ where $Q_B \in \mathbb{R}^{|T_c| \times r(B)}$. Moreover, let $W^\top = [I_{T_c}; C^\top]$ and $\Gamma = e_{k-|T|-r(B)}(W((Q_B^\perp)^\top Q_B^\perp)W^\top)$. Then the marginal probability of $T$ in DVS is*

$$P(T \subseteq S; A) = \frac{\left[\prod_{i=1}^{r(A_T)} \sigma_i^2(A_T)\right] \times \left[\prod_{j=1}^{r(B)} \sigma_j^2(B)\right] \times \Gamma}{Z_A}.$$

We prove Theorem 2 via a perturbation argument that connects DVS to volume sampling. Specifically, observe that for $\epsilon > 0$ and $|S| \geq n$ it holds that

$$\det(A_S A_S^\top + \varepsilon I_n) = \varepsilon^{n-k} \det(A_S^\top A_S + \varepsilon I_k) = \varepsilon^{n-k} \det\left(\begin{bmatrix} A_S \\ \sqrt{\varepsilon}(I_m)_S \end{bmatrix}^\top \begin{bmatrix} A_S \\ \sqrt{\varepsilon}(I_m)_S \end{bmatrix}\right). \quad (2.1)$$

Carefully letting $\epsilon \to 0$ bridges volumes with "dual" volumes. The technical remainder of the proof further relates this equality to singular values, and exploits properties of characteristic polynomials. A similar argument yields an alternative proof of Lemma 1. We show the proofs in detail in Appendix A and B respectively.

**Complexity.** The numerator of $P(T \subseteq S; A)$ in Theorem 2 requires $\mathcal{O}(mn^2)$ time to compute the first term, $\mathcal{O}(mn^2)$ to compute the second and $\mathcal{O}(m^3)$ to compute the third. The denominator takes $\mathcal{O}(mn^2)$ time, amounting in a total time of $\mathcal{O}(m^3)$ to compute the marginal probability.

## 2.2 Sampling

The marginal probabilities derived above directly yield a polynomial-time *exact* DVS algorithm. Instead of $k$-sets, we sample ordered $k$-tuples $\overrightarrow{S} = (s_1, \ldots, s_k) \in [m]^k$. We denote the $k$-tuple variant of the DVS distribution by $\overrightarrow{P}(\cdot; A)$:

$$\overrightarrow{P}((s_j = i_j)_{j=1}^k; A) = \frac{1}{k!} P(\{i_1, \ldots, i_k\}; A) = \prod_{j=1}^k \overrightarrow{P}(s_j = i_j | s_1 = i_1, \ldots, s_{j-1} = i_{j-1}; A).$$

Sampling $\overrightarrow{S}$ is now straightforward. At the $j$th step we sample $s_j$ via $\overrightarrow{P}(s_j = i_j | s_1 = i_1, \ldots, s_{j-1} = i_{j-1}; A)$; these probabilities are easily obtained from the marginals in Theorem 2.

**Corollary 3.** *Let $T = \{i_1, \ldots, i_{t-1}\}$, and $P(T \subseteq S; A)$ as in Theorem 2. Then,*

$$\overrightarrow{P}(s_t = i; A | s_1 = i_1, \ldots, s_{t-1} = i_{t-1}) = \frac{P(T \cup \{i\} \subseteq S; A)}{(k-t+1) \, P(T \subseteq S; A)}.$$

*As a result, it is possible to draw an exact dual volume sample in time $\mathcal{O}(km^4)$.*

The full proof may be found in the appendix. The running time claim follows since the sampling algorithm invokes $\mathcal{O}(mk)$ computations of marginal probabilities, each costing $\mathcal{O}(m^3)$ time.

**Remark** A potentially more efficient approximate algorithm could be derived by noting the relations between volume sampling and DVS. Specifically, we add a small perturbation to DVS as in Equation 2.1 to transform it into a volume sampling problem, and apply random projection for more efficient volume sampling as in [17]. Please refer to Appendix C for more details.

## 2.3 Derandomization

Next, we derandomize the above sampling algorithm to *deterministically* select a subset that satisfies the bound (1.3) for the Frobenius norm, thereby answering another question in [6]. The key insight for derandomization is that conditional expectations can be computed in polynomial time, given the marginals in Theorem 2:

**Corollary 4.** *Let* $(i_1, \ldots, i_{t-1}) \in [m]^{t-1}$ *be such that the marginal distribution satisfies* $\overrightarrow{P}(s_1 = i_1, \ldots, s_{t-1} = i_{t-1}; A) > 0$. *The conditional expectation can be expressed as*

$$\mathbb{E}\left[\|A_S^\dagger\|_F^2 \mid s_1 = i_1, \ldots, s_{t-1} = i_{t-1}\right] = \frac{\sum_{j=1}^n P'(\{i_1, \ldots, i_{t-1}\} \subseteq S \mid S \sim P(S; A_{[n]\setminus\{j\}}))}{P'(\{i_1, \ldots, i_{t-1}\} \subseteq S \mid S \sim P(S; A))},$$

*where* $P'$ *are the unnormalized marginal distributions, and it can be computed in* $\mathcal{O}(nm^3)$ *time.*

We show the full derivation in Appendix D.

Corollary 4 enables a greedy derandomization procedure. Starting with the empty tuple $\overrightarrow{S}_0 = \emptyset$, in the $i$th iteration, we greedily select $j^* \in \text{argmax}_j \mathbb{E}[\|A_{S \cup j}^\dagger\|_F^2 \mid (s_1, \ldots, s_i) = \overrightarrow{S}_{i-1} \circ j]$ and append it to our selection: $\overrightarrow{S}_i = \overrightarrow{S}_{i-1} \circ j$. The final set is the non-ordered version $S_k$ of $\overrightarrow{S}_k$. Theorem 5 shows that this greedy procedure succeeds, and implies a deterministic version of the bound (1.3).

**Theorem 5.** *The greedy derandomization selects a column set $S$ satisfying*

$$\|A_S^\dagger\|_F^2 \leq \frac{m-n+1}{k-n+1}\|A^\dagger\|_F^2; \quad \|A_S^\dagger\|_2^2 \leq \frac{n(m-n+1)}{k-n+1}\|A^\dagger\|_2^2.$$

In the proof, we construct a greedy algorithm. In each iteration, the algorithm computes, for each column that has not yet been selected, the expectation conditioned on this column being included in the current set. Then it chooses the element with the lowest conditional expectation to actually be added to the current set. This greedy inclusion of elements will only decrease the conditional expectation, thus retaining the bound in Theorem 5. The detailed proof is deferred to Appendix E.

**Complexity.** Each iteration of the greedy selection requires $\mathcal{O}(nm^3)$ to compute $\mathcal{O}(m)$ conditional expectations. Thus, the total running time for $k$ iterations is $\mathcal{O}(knm^4)$. The approximation bound for the spectral norm is slightly worse than that in (1.3), but is of the same order if $k = \mathcal{O}(n)$.

## 3 Strong Rayleigh Property and Fast Markov Chain Sampling

Next, we investigate DVS more deeply and discover that it possesses a remarkable structural property, namely, the *Strongly Rayleigh (SR)* [8] property. This property has proved remarkably fruitful in a variety of recent contexts, including recent progress in approximation algorithms [23], fast sampling [2, 27], graph sparsification [22, 39], extensions to the Kadison-Singer problem [1], and certain concentration of measure results [37], among others.

For DVS, the SR property has two major consequences: it leads to a fast mixing practical MCMC sampler, and it implies results on concentration of measure.

**Strongly Rayleigh measures.** SR measures were introduced in the landmark paper of Borcea et al. [8], who develop a rich theory of negatively associated measures. In particular, we say that a probability measure $\mu : 2^{[n]} \to \mathbf{R}_+$ is *negatively associated* if $\int F d\mu \int G d\mu \geq \int FG d\mu$ for $F, G$ increasing functions on $2^{[n]}$ with *disjoint* support. This property reflects a "repelling" nature of $\mu$, a property that occurs more broadly across probability, combinatorics, physics, and other fields—see [36, 8, 42] and references therein. The negative association property turns out to be quite subtle in general; the class of SR measures captures a strong notion of negative association and provides a framework for analyzing such measures.

Specifically, SR measures are defined via their connection to real stable polynomials [36, 8, 42]. A multivariate polynomial $f \in \mathbb{C}[z]$ where $z \in \mathbb{C}^m$ is called *real stable* if all its coefficients are real and $f(z) \neq 0$ whenever $\mathfrak{Im}(z_i) > 0$ for $1 \leq i \leq m$. A measure is called an *SR measure* if its multivariate generating polynomial $f_\mu(z) := \sum_{S \subseteq [n]} \mu(S) \prod_{i \in S} z_i$ is real stable. Notable examples of SR measures are Determinantal Point Processes [31, 29, 9, 26], balanced matroids [19, 37], Bernoullis conditioned on their sum, among others. It is known (see [8, pg. 523]) that the class of SR measures is exponentially larger than the class of determinantal measures.

## 3.1 Strong Rayleigh Property of DVS

Theorem 6 establishes the SR property for DVS and is the main result of this section. Here and in the following, we use the notation $z^S = \prod_{i \in S} z_i$.

**Theorem 6.** *Let $A \in \mathbb{R}^{n \times m}$ and $n \leq k \leq m$. Then the multiaffine polynomial*

$$p(z) := \sum_{|S|=k, S \subseteq [m]} \det(A_S A_S^\top) \prod_{i \in S} z_i \quad = \sum_{|S|=k, S \subseteq [m]} \det(A_S A_S^\top) z^S, \qquad (3.1)$$

*is real stable. Consequently, $P(S; A)$ is an SR measure.*

The proof of Theorem 6 relies on key properties of real stable polynomials and SR measures established in [8]. Essentially, the proof demonstrates that the generating polynomial of $\overline{P}(S_c; A)$ can be obtained by applying a few carefully chosen stability preserving operations to a polynomial that we know to be real stable. Stability, although easily destroyed, is closed under several operations noted in the important proposition below.

**Proposition 7** (Prop. 2.1 [8]). *Let $f : \mathbb{C}^m \to \mathbb{C}$ be a stable polynomial. The following properties preserve stability: (i) **Substitution**: $f(\mu, z_2, \ldots, z_m)$ for $\mu \in \mathbf{R}$; (ii) **Differentiation**: $\partial^S f(z_1, \ldots, z_m)$ for any $S \subseteq [m]$; (iii) **Diagonalization**: $f(z, z, z_3 \ldots, z_m)$ is stable, and hence $f(z, z, \ldots, z)$; and (iv) **Inversion**: $z_1 \cdots z_n f(z_1^{-1}, \ldots, z_n^{-1})$.*

In addition, we need the following two propositions for proving Theorem 6.

**Proposition 8** (Prop. 2.4 [7]). *Let $B$ be Hermitian, $z \in \mathbb{C}^m$ and $A_i$ ($1 \leq i \leq m$) be Hermitian semidefinite matrices. Then, the following polynomial is stable:*

$$f(z) := \det(B + \sum_i z_i A_i). \qquad (3.2)$$

**Proposition 9.** *For $n \leq |S| \leq m$ and $L := A^\top A$, we have $\det(A_S A_S^\top) = e_n(L_{S,S})$.*

*Proof.* Let $Y = \text{Diag}([y_i]_{i=1}^m)$ be a diagonal matrix. Using the Cauchy-Binet identity we have

$$\det(AYA^\top) = \sum_{|T|=n, T \subseteq [m]} \det((AY)_{:,T}) \det((A^\top)_{T,:}) = \sum_{|T|=n, T \subseteq [m]} \det(A_T^\top A_T) y^T.$$

Thus, when $Y = I_S$, the (diagonal) indicator matrix for $S$, we obtain $AYA^\top = A_S A_S^\top$. Consequently, in the summation above only terms with $T \subseteq S$ survive, yielding

$$\det(A_S A_S^\top) = \sum_{|T|=n, T \subseteq S} \det(A_T^\top A_T) = \sum_{|T|=n, T \subseteq S} \det(L_{T,T}) = e_n(L_{S,S}). \qquad \square$$

We are now ready to sketch the proof of Theorem 6.

*Proof. (Theorem 6).* Notationally, it is more convenient to prove that the "complement" polynomial $p_c(z) := \sum_{|S|=k, S \subseteq [m]} \det(A_S A_S^\top) z^{S_c}$ is stable; subsequently, an application of Prop. 7-(iv) yields stability of (3.1). Using matrix notation $W = \text{Diag}(w_1, \ldots, w_m)$, $Z = \text{Diag}(z_1, \ldots, z_m)$, our starting stable polynomial (this stability follows from Prop. 8) is

$$h(z, w) := \det(L + W + Z), \quad w \in \mathbb{C}^m, \; z \in \mathbb{C}^m,$$

which can be expanded as

$$h(z, w) = \sum_{S \subseteq [m]} \det(W_S + L_S) z^{S_c} = \sum_{S \subseteq [m]} \left( \sum_{T \subseteq S} w^{S \setminus T} \det(L_{T,T}) \right) z^{S_c}.$$

Thus, $h(z, w)$ is real stable in $2m$ variables, indexed below by $S$ and $R$ where $R := S \setminus T$. Instead of the form above, We can sum over $S, R \subseteq [m]$ but then have to constrain the support to the case when $S_c \cap T = \emptyset$ and $S_c \cap R = \emptyset$. In other words, we may write (using Iverson-brackets $[\![\cdot]\!]$)

$$h(z, w) = \sum_{S,R \subseteq [m]} [\![S_c \cap R = \emptyset \wedge S_c \cap T = \emptyset]\!] \det(L_{T,T}) z^{S_c} w^R. \qquad (3.3)$$

Next, we truncate polynomial (3.3) at degree $(m-k)+(k-n) = m-n$ by restricting $|S_c \cup R| = m-n$. By [8, Corollary 4.18] this truncation preserves stability, whence

$$H(z, w) := \sum_{\substack{S,R \subseteq [m] \\ |S_c \cup R| = m-n}} [\![S_c \cap R = \emptyset]\!] \det(L_{S \setminus R, S \setminus R}) z^{S_c} w^R,$$

is also stable. Using Prop. 7-(iii), setting $w_1 = \ldots = w_m = y$ retains stability; thus

$$g(z, y) := H(z, (\underbrace{y, y, \ldots, y}_{m \text{ times}})) = \sum_{\substack{S,R \subseteq [m] \\ |S_c \cup R| = m-n}} [\![S_c \cap R = \emptyset]\!] \det(L_{S \setminus R, S \setminus R}) z^{S_c} y^{|R|}$$

$$= \sum_{S \subseteq [m]} \left( \sum_{|T|=n, T \subseteq S} \det(L_{T,T}) \right) y^{|S|-|T|} z^{S_c} = \sum_{S \subseteq [m]} e_n(L_{S,S}) y^{|S|-n} z^{S_c},$$

is also stable. Next, differentiating $g(z, y)$, $k - n$ times with respect to $y$ and evaluating at $0$ preserves stability (Prop. 7-(ii) and (i)). In doing so, only terms corresponding to $|S| = k$ survive, resulting in

$$\left. \frac{\partial^{k-n}}{\partial y^{k-n}} g(z, y) \right|_{y=0} = (k-n)! \sum_{|S|=k, S \subseteq [m]} e_n(L_{S,S}) z^{S_c} = (k-n)! \sum_{|S|=k, S \subseteq [m]} \det(A_S A_S^\top) z^{S_c},$$

which is just $p_c(z)$ (up to a constant); here, the last equality follows from Prop. 9. This establishes stability of $p_c(z)$ and hence of $p(z)$. Since $p(z)$ is in addition multiaffine, it is the generating polynomial of an SR measure, completing the proof. $\qquad \square$

## 3.2 Implications: MCMC

The SR property of $P(S; A)$ established in Theorem 6 implies a fast mixing Markov chain for sampling $S$. The states for the Markov chain are all sets of cardinality $k$. The chain starts with a randomly-initialized active set $S$, and in each iteration we swap an element $s^{\text{in}} \in S$ with an element $s^{\text{out}} \notin S$ with a specific probability determined by the probability of the current and proposed set. The stationary distribution of this chain is the one induced by DVS, by a simple detailed-balance argument. The chain is shown in Algorithm 1.

---

**Algorithm 1** Markov Chain for Dual Volume Sampling

**Input:** $A \in \mathbb{R}^{n \times m}$ the matrix of interest, $k$ the target cardinality, $T$ the number of steps
**Output:** $S \sim P(S; A)$
Initialize $S \subseteq [m]$ such that $|S| = k$ and $\det(A_S A_S^\top) > 0$
**for** $i = 1$ to $T$ **do**
    draw $b \in \{0, 1\}$ uniformly
    **if** $b = 1$ **then**
        Pick $s^{\text{in}} \in S$ and $s^{\text{out}} \in [m] \setminus S$ uniformly randomly
        $q(s^{\text{in}}, s^{\text{out}}, S) \leftarrow \min \left\{ 1, \det(A_{S \cup \{s^{\text{out}}\} \setminus \{s^{\text{in}}\}} A_{S \cup \{s^{\text{out}}\} \setminus \{s^{\text{in}}\}}^\top) / \det(A_S A_S^\top) \right\}$
        $S \leftarrow S \cup \{s^{\text{out}}\} \setminus \{s^{\text{in}}\}$ with probability $q(s^{\text{in}}, s^{\text{out}}, S)$
    **end if**
**end for**

---

The convergence of the markov chain is measured via its mixing time: The *mixing time* of the chain indicates the number of iterations $t$ that we must perform (starting from $S_0$) before we can consider $S_t$ as an approximately valid sample from $P(S; A)$. Formally, if $\delta_{S_0}(t)$ is the total variation distance between the distribution of $S_t$ and $P(S; A)$ after $t$ steps, then

$$\tau_{S_0}(\varepsilon) := \min\{t : \delta_{S_0}(t') \leq \varepsilon, \ \forall t' \geq t\}$$

is the *mixing time* to sample from a distribution $\varepsilon$-close to $P(S; A)$ in terms of total variation distance. We say that the chain mixes fast if $\tau_{S_0}$ is polynomial in the problem size.

The fast mixing result for Algorithm 1 is a corollary of Theorem 6 combined with a recent result of [3] on fast-mixing Markov chains for homogeneous SR measures. Theorem 10 states this precisely.

**Theorem 10** (Mixing time). *The mixing time of Markov chain shown in Algorithm 1 is given by*

$$\tau_{S_0}(\varepsilon) \leq 2k(m-k)(\log P(S_0; A)^{-1} + \log \varepsilon^{-1}).$$

*Proof.* Since $P(S; A)$ is $k$-homogeneous SR by Theorem 6, the chain constructed for sampling $S$ following that in [3] mixes in $\tau_{S_0}(\varepsilon) \leq 2k(m-k)(\log P(S_0; A)^{-1} + \log \varepsilon^{-1})$ time. $\square$

**Implementation.** To implement Algorithm 1 we need to compute the transition probabilities $q(s^{\text{in}}, s^{\text{out}}, S)$. Let $T = S \backslash \{s^{\text{in}}\}$ and assume $r(A_T) = n$. By the matrix determinant lemma we have the acceptance ratio

$$\frac{\det(A_{S \cup \{s^{\text{out}}\} \backslash \{s^{\text{in}}\}} A_{S \cup \{s^{\text{out}}\} \backslash \{s^{\text{in}}\}}^\top)}{\det(A_S A_S^\top)} = \frac{(1 + A_{\{s^{\text{out}}\}}^\top (A_T A_T^\top)^{-1} A_{\{s^{\text{out}}\}})}{(1 + A_{\{s^{\text{in}}\}}^\top (A_T A_T^\top)^{-1} A_{\{s^{\text{in}}\}})}.$$

Thus, the transition probabilities can be computed in $\mathcal{O}(n^2 k)$ time. Moreover, one can further accelerate this algorithm by using the quadrature techniques of [28] to compute lower and upper bounds on this acceptance ratio to determine early acceptance or rejection of the proposed move.

**Initialization.** A remaining question is initialization. Since the mixing time involves $\log P(S_0; A)^{-1}$, we need to start with $S_0$ such that $P(S_0; A)$ is sufficiently bounded away from 0. We show in Appendix F that by a simple greedy algorithm, we are able to initialize $S$ such that $\log P(S; A)^{-1} \geq \log(2^n k! \binom{m}{k}) = \mathcal{O}(k \log m)$, and the resulting running time for Algorithm 1 is $\widetilde{\mathcal{O}}(k^3 n^2 m)$, which is *linear* in the size of data set $m$ and is efficient when $k$ is not too large.

### 3.3 Further implications and connections

**Concentration.** Pemantle and Peres [37] show concentration results for strong Rayleigh measures. As a corollary of our Theorem 6 together with their results, we directly obtain tail bounds for DVS.

**Algorithms for experimental design.** Widely used, classical algorithms for finding an approximate optimal design include Fedorov's exchange algorithm [20, 21] (a greedy local search) and simulated annealing [34]. Both methods start with a random initial set $S$, and greedily or randomly exchange a column $i \in S$ with a column $j \notin S$. Apart from very expensive running times, they are known to work well in practice [35, 43]. Yet so far there is no theoretical analysis, or a principled way of determining when to stop the greedy search.

Curiously, our MCMC sampler is essentially a randomized version of Fedorov's exchange method. The two methods can be connected by a unified, simulated annealing view, where we define $P^\beta(S; A) \propto \exp\{\log \det(A_S A_S^\top)/\beta\}$ with temperature parameter $\beta$. Driving $\beta$ to zero essentially recovers Fedorov's method, while our results imply fast mixing for $\beta = 1$, together with approximation guarantees. Through this lens, simulated annealing may be viewed as initializing Fedorov's method with the fast-mixing sampler. In practice, we observe that letting $\beta < 1$ improves the approximation results, which opens interesting questions for future work.

## 4 Experiments

We report selection performance of DVS on real regression data (CompAct, CompAct(s), Abalone and Bank32NH[1]) for experimental design. We use 4,000 samples from each dataset for estimation. We compare against various baselines, including uniform sampling (Unif), leverage score sampling (Lev) [30], predictive length sampling (PL) [45], the sampling (Smpl)/greedy (Greedy) selection methods in [43] and Fedorov's exchange algorithm [20]. We initialize the MCMC sampler with Kmeans++ [5] for DVS and run for 10,000 iterations, which empirically yields selections that are

sufficiently good. We measure performances via (1) the prediction error $\|y - X\hat{\alpha}\|$, and 2) running times. Figure 1 shows the results for these three measures with sample sizes $k$ varying from 60 to 200. Further experiments (including for the interpolation $\beta < 1$), may be found in the appendix.

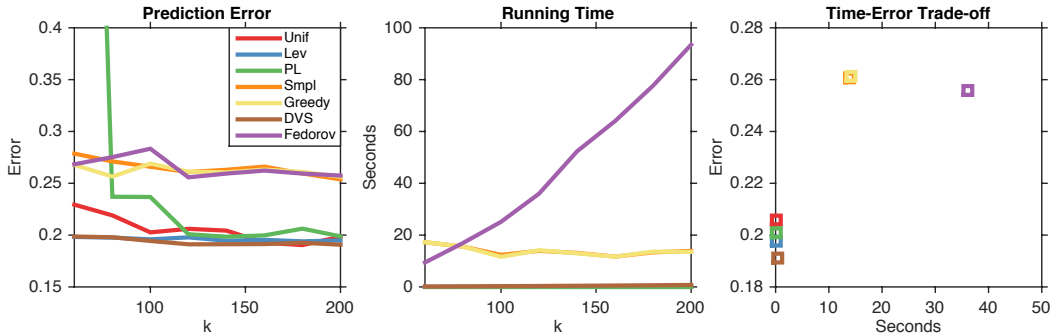

Figure 1: Results on the CompAct(s) dataset. Results are the median of 10 runs, except `Greedy` and `Fedorov`. Note that `Unif`, `Lev`, `PL` and `DVS` use less than 1 second to finish experiments.

In terms of prediction error, DVS performs well and is comparable with `Lev`. Its strength compared to the greedy and relaxation methods (`Smpl`, `Greedy`, `Fedorov`) is running time, leading to good time-error tradeoffs. These tradeoffs are illustrated in Figure 1 for $k = 120$.

In other experiments (shown in Appendix G) we observed that in some cases, the optimization and greedy methods (`Smpl`, `Greedy`, `Fedorov`) yield better results than sampling, however with much higher running times. Hence, given time-error tradeoffs, DVS may be an interesting alternative in situations where time is a very limited resource and results are needed quickly.

## 5   Conclusion

In this paper, we study the problem of DVS and develop an exact (randomized) polynomial time sampling algorithm as well as its derandomization. We further study dual volume sampling via the theory of real-stable polynomials and prove that its distribution satisfies the "Strong Rayleigh" property. This result has remarkable consequences, especially because it implies a provably fast-mixing Markov chain sampler that makes dual volume sampling much more attractive to practitioners. Finally, we observe connections to classical, computationally more expensive experimental design methods (Fedorov's method and SA); together with our results here, these could be a first step towards a better theoretical understanding of those methods.

**Acknowledgement**

This research was supported by NSF CAREER award 1553284, NSF grant IIS-1409802, DARPA grant N66001-17-1-4039, DARPA FunLoL grant (W911NF-16-1-0551) and a Siebel Scholar Fellowship. The views, opinions, and/or findings contained in this article are those of the author and should not be interpreted as representing the official views or policies, either expressed or implied, of the Defense Advanced Research Projects Agency or the Department of Defense.

## Footnotes

[1] http://www.dcc.fc.up.pt/?ltorgo/Regression/DataSets.html

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
