[Supplementary Material]

# A   Partition Function

We recall two easily verified facts about determinants that will be useful in our analysis:

$$\det(K + uv^\top) = \det(K)(1 + u^\top K^{-1}v), \quad \text{for } K \in \mathrm{GL}_n(\mathbb{R}), \tag{A.1}$$

$$a^{m-n}\det(AA^\top + aI_n) = \det(A^\top A + aI_m), \quad \text{for } A \in \mathbb{R}^{n \times m} \ (n \le m), \text{ and } a > 0. \tag{A.2}$$

The first one is known as matrix determinant lemma.

The partition function of $P(\cdot; A)$, happens to have a pleasant closed-form formula. Although this formula is known [6], and follows immediately by an application of the Cauchy-Binet identity, we present an alternative proof based on the perturbation argument for its conceptual value and subsequent use.

**Theorem 11** (Partition Function [6]). *Given $A \in \mathbb{R}^{n \times m}$ where $r(A) = n$ and $n \le |S| = k \le m$, we have*

$$\sum_{|S|=k, S \subseteq [m]} \det(A_S A_S^\top) = \binom{m-n}{k-n} \det(AA^\top). \tag{A.3}$$

*Proof.* First note that for $n \le |S| = k \le m$ and any $\varepsilon > 0$, by (A.2) we have

$$\det(A_S A_S^\top + \varepsilon I_n) = \frac{1}{\varepsilon^{k-n}} \det(A_S^\top A_S + \varepsilon I_k)$$

Taking limits as $\varepsilon \to 0$ on both sides we have

$$\det(A_S A_S^\top) = \lim_{\varepsilon \to 0} \det(A_S A_S^\top + \varepsilon I_n) = \lim_{\varepsilon \to 0} \frac{1}{\varepsilon^{k-n}} \det(A_S^\top A_S + \varepsilon I_k).$$

Let us focus on $\det(A_S^\top A_S + \varepsilon I_k)$. We construct an identity matrix $I_m \in \mathbb{R}^{m \times m}$, then we have

$$\det(A_S^\top A_S + \varepsilon I_k) = \det(A_S^\top A_S + \varepsilon I_S^\top I_S) = \det(A_S^\top A_S + (\sqrt{\varepsilon} I_S)^\top \sqrt{\varepsilon} I_S)$$

$$= \det\left( \begin{bmatrix} A_S \\ \sqrt{\varepsilon}(I_m)_S \end{bmatrix}^\top \begin{bmatrix} A_S \\ \sqrt{\varepsilon}(I_m)_S \end{bmatrix} \right) \propto \widehat{P}\left( S; \begin{bmatrix} A \\ \sqrt{\varepsilon}I_m \end{bmatrix} \right). \tag{A.4}$$

In other words, this value is proportional to the probability of sampling columns from $\begin{bmatrix} A \\ \sqrt{\varepsilon}I_m \end{bmatrix}$ using volume sampling. Therefore, using the definition of $e_k$ we have

$$\frac{1}{\varepsilon^{k-n}} \sum_{|S|=k, S \subseteq [m]} \det(A_S^\top A_S + \varepsilon I_k) = \frac{1}{\varepsilon^{k-n}} e_k(A^\top A + \varepsilon I_m)$$

$$= \frac{1}{\varepsilon^{k-n}} e_k(\mathrm{Diag}([(\sigma_1^2(A) + \varepsilon), (\sigma_2^2(A) + \varepsilon), \ldots, (\sigma_n^2(A) + \varepsilon), \varepsilon, \ldots, \varepsilon]))$$

$$= \binom{m-n}{k-n} \prod_{i=1}^n (\sigma_i^2(A) + \varepsilon) + O(\varepsilon).$$

Now taking the limit as $\varepsilon \to 0$ we obtain

$$\sum_{|S|=k, S \subseteq [m]} \det(A_S A_S^\top) = \lim_{\varepsilon \to 0} \binom{m-n}{k-n} \prod_{i=1}^n (\sigma_i^2(A) + \varepsilon) + O(\varepsilon) = \binom{m-n}{k-n} \det(AA^\top).$$

$\square$

# B   Marginal Probability

*Proof.* The marginal probability of a set $T \subseteq [m]$ for dual volume sampling is

$$P(T \subseteq S; A) = \frac{\sum_{S \supseteq T, |S|=k} \det(A_S A_S^\top)}{\sum_{|S'|=k} \det(A_{S'} A_{S'}^\top)}.$$

Theorem 11 shows how to compute the denominator, thus our main effort is devoted to the nominator. We have

$$\sum_{S \supseteq T, |S|=k} \det(A_S A_S^\top) = \sum_{R \cap T = \emptyset, |R|=k-|T|} \det(A_{T \cup R} A_{T \cup R}^\top)$$

Using the $\varepsilon$-trick we have

$$\sum_{R \cap T = \emptyset, |R|=k-|T|} \det(A_{T \cup R} A_{T \cup R}^\top) = \lim_{\varepsilon \to 0} \sum_{R \cap T = \emptyset, |R|=k-|T|} \det(A_{T \cup R} A_{T \cup R}^\top + \varepsilon I_n)$$

$$= \lim_{\varepsilon \to 0} \frac{1}{\varepsilon^{k-n}} \sum_{R \cap T = \emptyset, |R|=k-|T|} \det(A_{T \cup R}^\top A_{T \cup R} + \varepsilon I_k).$$

By decomposing $\det(A_{T \cup R}^\top A_{T \cup R} + \varepsilon I_k)$ we have

$$\det(A_{T \cup R}^\top A_{T \cup R} + \varepsilon I_k)$$
$$= \det(A_T^\top A_T + \varepsilon I_{|T|}) \det\left(A_R^\top A_R + \varepsilon I_{|R|} - A_R^\top A_T (A_T^\top A_T + \varepsilon I_{|T|})^{-1} A_T^\top A_R\right).$$

Now we let $A_T = Q_T \Sigma_T V_T^\top$ be the singular value decomposition of $A_T$ where $Q_T \in \mathbb{R}^{n \times r(A_T)}$, $\Sigma_T \in \mathbb{R}^{r(A_T) \times |T|}$ and $V_T \in \mathbb{R}^{|T| \times |T|}$. Plugging the decomposition in the equation we obtain

$$A_R^\top A_T (A_T^\top A_T + \varepsilon I_{|T|})^{-1} A_T^\top A_R = A_R^\top Q_T \Sigma_T V_T^\top (V_T \Sigma_T^\top \Sigma_T V_T^\top + \varepsilon I_{|T|})^{-1} V_T \Sigma_T^\top Q_T^\top A_R$$

$$= A_R^\top Q_T \Sigma_T (\Sigma_T^\top \Sigma_T + \varepsilon I_{|T|})^{-1} \Sigma_T^\top Q_T^\top A_R$$

$$= A_R^\top Q_T \begin{bmatrix} \frac{\sigma_1^2(A_T)}{\sigma_1^2(A_T)+\varepsilon} & 0 & \cdots & 0 \\ 0 & \frac{\sigma_2^2(A_T)}{\sigma_2^2(A_T)+\varepsilon} & \cdots & 0 \\ \vdots & \vdots & \ddots & \vdots \\ 0 & 0 & \cdots & \frac{\sigma_{r(A_T)}^2(A_T)}{\sigma_{r(A_T)}^2(A_T)+\varepsilon} \end{bmatrix} Q_T^\top A_R$$

$$= A_R^\top Q_T Q_T^\top A_R - \varepsilon A_R^\top Q_T \begin{bmatrix} \frac{1}{\sigma_1^2(A_T)+\varepsilon} & 0 & \cdots & 0 \\ 0 & \frac{1}{\sigma_2^2(A_T)+\varepsilon} & \cdots & 0 \\ \vdots & \vdots & \ddots & \vdots \\ 0 & 0 & \cdots & \frac{1}{\sigma_{r(A_T)}^2(A_T)+\varepsilon} \end{bmatrix} Q_T^\top A_R.$$

Thus it follows that

$$A_R^\top A_R + \varepsilon I_{|R|} - A_R^\top A_T (A_T^\top A_T + \varepsilon I_{|T|})^{-1} A_T^\top A_R$$

$$= A_R^\top (I - Q_T Q_T^\top) A_R + \varepsilon A_R^\top Q_T \begin{bmatrix} \frac{1}{\sigma_1^2(A_T)+\varepsilon} & 0 & \cdots \\ 0 & \frac{1}{\sigma_2^2(A_T)+\varepsilon} & \cdots \\ \vdots & \vdots & \ddots \end{bmatrix} Q_T^\top A_R + \varepsilon I_{|R|}$$

$$= B_R^\top B_R + \varepsilon C_R^\top C_R + \varepsilon I_{|R|},$$

where $B_R$ is the projection of columns of $A_R$ on the orthogonal space of columns of $A_T$. Let $Q_T^\perp \in \mathbb{R}^{n \times (n-r(A_T))}$ be the complement column space of $Q_T$, then we have $B_R = (Q_T^\perp)^\top A_R \in \mathbb{R}^{(n-r(A_T)) \times |R|}$. Moreover,

$$C_R = \begin{bmatrix} \frac{1}{\sqrt{\sigma_1^2(A_T)+\varepsilon}} & 0 & \cdots \\ 0 & \frac{1}{\sqrt{\sigma_2^2(A_T)+\varepsilon}} & \cdots \\ \vdots & \vdots & \ddots \end{bmatrix} Q_T^\top A_R \in \mathbb{R}^{r(A_T) \times |R|}.$$

We further let $B_{T_c} = (Q_T^\perp)^\top A_{T_c} \in \mathbb{R}^{(n-r(A_T)) \times (m-|T|)}$ and

$$C_{T_c} = \begin{bmatrix} \frac{1}{\sqrt{\sigma_1^2(A_T)+\varepsilon}} & 0 & \cdots \\ 0 & \frac{1}{\sqrt{\sigma_2^2(A_T)+\varepsilon}} & \cdots \\ \vdots & \vdots & \ddots \end{bmatrix} Q_T^\top A_{T_c} \in \mathbb{R}^{r(A_T) \times (m-|T|)}$$

where $T_c = [m]\backslash T$. Then we have

$$\sum_{R\cap T=\emptyset, |R|=k-|T|} \det(A_{T\cup R}^\top A_{T\cup R} + \varepsilon I_k)$$

$$= \det(A_T^\top A_T + \varepsilon I_{|T|}) \sum_{R\cap T=\emptyset, |R|=k-|T|} \det(B_R^\top B_R + \varepsilon C_R^\top C_R + \varepsilon I_{|R|})$$

$$= \det(A_T^\top A_T + \varepsilon I_{|T|}) \times e_{k-|T|} \left( \begin{bmatrix} B_{T_c} \\ \sqrt{\varepsilon} U_{T_c} \\ \sqrt{\varepsilon} C_{T_c} \end{bmatrix} \begin{bmatrix} B_{T_c} \\ \sqrt{\varepsilon} U_{T_c} \\ \sqrt{\varepsilon} C_{T_c} \end{bmatrix}^\top \right)$$

where we construct an orthonormal matrix $U \in \mathbb{R}^{(m-|T|)\times(m-|T|)}$ whose columns are basis vectors. Since we are free to chose any orthonormal $U$, we simply let it be $I$. Let $W_{T_c} = \begin{bmatrix} I_{T_c} \\ C_{T_c} \end{bmatrix}$, we have

$$\left( \begin{bmatrix} B_{T_c} \\ \sqrt{\varepsilon} U_{T_c} \\ \sqrt{\varepsilon} C_{T_c} \end{bmatrix} \begin{bmatrix} B_{T_c} \\ \sqrt{\varepsilon} U_{T_c} \\ \sqrt{\varepsilon} C_{T_c} \end{bmatrix}^\top \right) = \left( \begin{bmatrix} B_{T_c} \\ \sqrt{\varepsilon} W_{T_c} \end{bmatrix} \begin{bmatrix} B_{T_c} \\ \sqrt{\varepsilon} W_{T_c} \end{bmatrix}^\top \right)$$

$$= F_{T_c} \in \mathbb{R}^{(m+n-|T|)\times(m+n-|T|)}$$

The properties of characteristic polynomials imply that

$$e_{k-|T|}(F_{T_c}) = \sum_{|S|=k-|T|} \det((F_{T_c})_{S,S})$$

$$= \sum_{S_1,S_2} \det((F_{T_c})_{S_1,S_1}) \det((F_{T_c})_{S_2,S_2} - (F_{T_c})_{S_2,S_1}(F_{T_c})_{S_1,S_1}^{-1}(F_{T_c})_{S_1,S_2})$$

where $S_1 = S \cap [r(B_{T_c})]$ and $S_2 = [m+n-|T|]\backslash S_1$. Further we have

$$\sum_{S_1,S_2} \det((F_{T_c})_{S_1,S_1}) \det((F_{T_c})_{S_2,S_2} - (F_{T_c})_{S_2,S_1}(F_{T_c})_{S_1,S_1}^{-1}(F_{T_c})_{S_1,S_2})$$

$$= \sum_{S_1,S_2} \varepsilon^{k-|T|-|S_1|} \det((B_{T_c})_{S_1}(B_{T_c})_{S_1}^\top) \times$$

$$\det((W_{T_c})_{S_2}(W_{T_c})_{S_2}^\top - (W_{T_c})_{S_2}(B_{T_c})_{S_1}^\top((B_{T_c})_{S_1}(B_{T_c})_{S_1}^\top)^{-1}(B_{T_c})_{S_1}(W_{T_c})_{S_2}^\top)$$

Hence it follows that

$$\lim_{\varepsilon\to 0}\frac{1}{\varepsilon^{k-n}} \sum_{R\cap T=\emptyset,|R|=k-|T|} \det(A_{T\cup R}^\top A_{T\cup R} + \varepsilon I_k) = \lim_{\varepsilon\to 0}\frac{1}{\varepsilon^{k-n}} \det(A_T^\top A_T + \varepsilon I_{|T|}) \times e_{k-|T|}(F_{T_c})$$

$$= \lim_{\varepsilon\to 0}\frac{1}{\varepsilon^{k-n}}\varepsilon^{|T|-r(A_T)} \left[ \prod_{i=1}^{r(A_T)}(\sigma_i^2(A_T)+\varepsilon) \right] \times$$

$$\sum_{|S|=k-|T|} \varepsilon^{k-|T|-|S_1|} \det((B_{T_c})_{S_1}(B_{T_c})_{S_1}^\top) \times$$

$$\det((W_{T_c})_{S_2}(W_{T_c})_{S_2}^\top - (W_{T_c})_{S_2}(B_{T_c})_{S_1}^\top((B_{T_c})_{S_1}(B_{T_c})_{S_1}^\top)^{-1}(B_{T_c})_{S_1}(W_{T_c})_{S_2}^\top)$$

(Since $r(A_T) + r(B_{T_c}) = n$ and $|S_1| \le r(B_{T_c})$)

$$= \lim_{\varepsilon\to 0}\frac{1}{\varepsilon^{k-n}}\varepsilon^{|T|-r(A_T)} \left[ \prod_{i=1}^{r(A_T)}(\sigma_i^2(A_T)+\varepsilon) \right] \times$$

$$\sum_{|S|=k-|T|} \varepsilon^{k-|T|-r(B_{T_c})} \det(B_{T_c}B_{T_c}^\top) \det((W_{T_c})_{S_2}(W_{T_c})_{S_2}^\top - (W_{T_c})_{S_2}B_{T_c}^\top(B_{T_c}B_{T_c}^\top)^{-1}B_{T_c}(W_{T_c})_{S_2}^\top) + O(\varepsilon)$$

$$= \left[ \prod_{i=1}^{r(A_T)}\sigma_i^2(A_T) \right] \times \left[ \prod_{j=1}^{r(B_{T_c})}\sigma_j^2(B_{T_c}) \right] \sum_{S_2} \det((W_{T_c})_{S_2}(W_{T_c})_{S_2}^\top - (W_{T_c})_{S_2}B_{T_c}^\top(B_{T_c}B_{T_c}^\top)^{-1}B_{T_c}(W_{T_c})_{S_2}^\top)$$

where $S_2 \subseteq [m + n - |T|]\setminus[r(B_{T_c})]$ and $|S_2| = k - |T| - r(B_{T_c})$.

Let $Q_{B_{T_c}} \operatorname{diag}(\sigma_i^2(B_{T_c}))Q_{B_{T_c}}^\top$ be the eigenvalue decomposition of $B_{T_c}^\top B_{T_c}$ where $Q_{B_{T_c}} \in \mathbb{R}^{|T_c| \times r(B_{T_c})}$. Further, let $Q_{B_{T_c}}^\perp$ be the complement column space of $Q_{B_{T_c}}$, thus we have

$$\begin{bmatrix} Q_{B_{T_c}}^\top \\ (Q_{B_{T_c}}^\perp)^\top \end{bmatrix} \begin{bmatrix} Q_{B_{T_c}} & Q_{B_{T_c}}^\perp \end{bmatrix} = I_{|T_c|} = I_{n-|T|}$$

Then for any $S_2 \subseteq [m + n - |T|]\setminus[r(B_{T_c})]$ we have

$$\det((W_{T_c})_{S_2}(W_{T_c})_{S_2}^\top - (W_{T_c})_{S_2}B_{T_c}^\top(B_{T_c}B_{T_c}^\top)^{-1}B_{T_c}(W_{T_c})_{S_2}^\top) = \det(W_{S_2}(I_{n-|T|} - Q_{B_{T_c}}Q_{B_{T_c}}^\top)(W_{T_c})_{S_2}^\top)$$
$$= \det((W_{T_c})_{S_2}(Q_{B_{T_c}}^\perp(Q_{B_{T_c}}^\perp)^\top)(W_{T_c})_{S_2}^\top)$$

It follows that

$$\sum_{S_2} \det(W_{S_2}(W_{T_c})_{S_2}^\top - (W_{T_c})_{S_2}B_{T_c}^\top(B_{T_c}B_{T_c}^\top)^{-1}B_{T_c}(W_{T_c})_{S_2}^\top) = e_{k-|T|-r(B_{T_c})}(W_{T_c}((Q_{B_{T_c}}^\perp)^\top Q_{B_{T_c}}^\perp)W_{T_c}^\top)$$

$$= E_T$$

Combining all the above derivations, we obtain that

$$\Pr(T \subseteq S | S \sim P(S; A)) = \frac{\left[\prod_{i=1}^{r(A_T)} \sigma_i^2(A_T)\right] \times \left[\prod_{j=1}^{r(B_{T_c})} \sigma_j^2(B_{T_c})\right] \times \Gamma_T}{\binom{n-m}{k-m} \det(AA^\top)}.$$

$\square$

## C  Approximate Sampling via Volume Sampling

**Corollary 12** (Approximate DVS via Random Projection). *For any $\varepsilon > 0$ and $\delta_2 > 0$ there is an algorithm that, in time $\widetilde{\mathcal{O}}(\frac{k^2 nm}{\delta_2^2} + \frac{k^7 m}{\delta_2^6})$, samples a subset from an approximate distribution $\widetilde{P}(\cdot; A)$ with $\delta_1 = \max_{|S|=k}(1 + \frac{\varepsilon}{\sigma_{\min}^2(A_S)})^n - 1 \approx \frac{n\varepsilon}{\sigma_{\min}^2(A_S)}$ and*

$$\frac{\widetilde{P}(S; A)}{(1+\delta_1)(1+\delta_2)} \le P(S; A) \le (1+\delta_1)(1+\delta_2)\widetilde{P}(S; A); \quad \forall S \subseteq [m].$$

It may happen in practice that $n \ll m$ but $k$ is of the same order as $n$. In such case we can transform the dual volume sampling to slightly distorted volume sampling based on (A.2) and then take the advantage of determinant-preserving projections to accelerate the sampling procedure.

Concretely, instead of sampling column subset $S$ with probability proportional to $\det(A_S A_S^\top)$, we sample with probability proportional to a distorted value $\det(A_S A_S^\top + \varepsilon I_n)$ for small $\varepsilon > 0$. Denoting this distorted distribution as $P_\varepsilon(S; A)$, we have

$$P_\varepsilon(S; A) = \frac{1}{\varepsilon^{k-n}} \det(A_S^\top A_S + \varepsilon I_k) = \frac{1}{\varepsilon^{k-n}} \prod_{i=1}^{n} (\sigma_i^2(A_S) + \varepsilon).$$

Letting $\sigma_{\min}(A_S) > 0$ be the minimum singular value, we have

$$1 \le \frac{\prod_{i=1}^{n}(\sigma_i^2(A_S) + \varepsilon)}{\prod_{i=1}^{n}(\sigma_i^2(A_S))} \le (1 + \frac{\varepsilon}{\sigma_{\min}^2(A_S)})^n.$$

We further let

$$\delta_1 = \max_{|S|=k}(1 + \frac{\varepsilon}{\sigma_{\min}^2(A_S)})^n - 1 \approx \frac{n\varepsilon}{\sigma_{\min}^2(A_S)},$$

when $\varepsilon$ sufficiently small. Sampling from $P_\varepsilon$ will yield $(1 + \delta_1)$-approximate dual volume sampling (in the sense of [17] and our Theorem 12). We can sample from $P_\varepsilon$ via *volume sampling* with distribution $\widehat{P}(S; \begin{bmatrix} A \\ \sqrt{\varepsilon}I_m \end{bmatrix})$. With the volume sampling algorithm proposed in [17], the resulting running time would be $\widetilde{\mathcal{O}}(km^4)$.

To accelerate sampling procedure, we consider random projection techniques that preserve volumes. [32] showed that Gaussian random projections indeed preserve volumes as we need:

**Theorem 13** (Random Projection [32]). *For any $X \in \mathbb{R}^{n \times m}$, $1 \leq k \leq m$ and $0 < \delta_2 \leq 1/2$, the random Gaussian projection of $\mathbb{R}^m \to \mathbb{R}^d$ where*

$$d = \mathcal{O}\left(\frac{k^2 \log n}{\delta_2^2}\right),$$

*satisfies*

$$\det(X_S^\top X_S) \leq \det(\widetilde{X}_S^\top \widetilde{X}_S) \leq (1 + \delta_2) \det(X_S^\top X_S) \tag{C.1}$$

*for all $S \subseteq [n]$ and $|S| \leq k$ where $\widetilde{X}$ is the projected matrix.*

This theorem completes what we need to prove Corollary 12.

*Proof.* (Corollary 12) The idea is to project $\begin{bmatrix} A \\ \sqrt{\varepsilon} I_m \end{bmatrix}$ to a lower-dimensional space in a way that the values for submatrix determinants are preserved up to a small multiplicative factor. Then we perform volume sampling. We project columns of $\begin{bmatrix} A \\ \sqrt{\varepsilon} I_m \end{bmatrix}$, which is in $\mathbb{R}^{m+n}$, to vectors in $\mathbb{R}^d$ where $d = \mathcal{O}\left(\frac{k^2 \log m}{\delta_2^2}\right)$ so as to achieve a $(1 + \delta_2)$ approximation by Theorem 13. Let $G$ be a $d \times (m + n)$-dimensional i.i.d. Gaussian random matrix, then we have

$$G \begin{bmatrix} A \\ \sqrt{\varepsilon} I \end{bmatrix} = G_A A + \sqrt{\varepsilon} G'_A \tag{C.2}$$

where $G_A \in \mathbb{R}^{d \times n}$ and $G'_A \in \mathbb{R}^{d \times n}$ are two independent Gaussian random matrix. The projected matrix can be computed in $\mathcal{O}(dnm) = \widetilde{\mathcal{O}}(k^2 nmn/\delta_2^2)$ time. After that, if we use volume sampling algorithm proposed in [17] the resulting running time would be $\mathcal{O}(kd^3 m) = \widetilde{\mathcal{O}}(k^7 m/\delta_2^6)$. Thus the total running time would be $\widetilde{\mathcal{O}}(\frac{k^2 nm}{\delta_2^2} + \frac{k^7 m}{\delta_2^6})$. $\square$

**Remarks.** An interesting observation is that the resulting running time is independent of $\delta_1$, which means one can set $\varepsilon$ arbitrarily small so as to make the approximation in the first step as accurate as possible, without affecting the running time. However, in practice, a very small $\varepsilon$ can result in numerical problems. In addition, the dimensionality reduction is only efficient if $d < m + n$.

## D   Conditional Expectation

*Proof.* We use $A^j$ denote the matrix $A_{[n] \setminus \{j\}, :}$, namely matrix $A$ with row $j$ deleted. We have

$$\mathbb{E}\left[\|A_S^\dagger\|_F^2 \mid s_1 = i_1, \ldots, s_{t-1} = i_{t-1}\right]$$

$$= \sum_{(i_t, \ldots, i_k) \in [m]^{k-t+1}} \|A_S^\dagger\|_F^2 \overrightarrow{P}(s_1 = i_1, \ldots, s_k = i_k; A \mid s_1 = i_1, \ldots, s_{t-1} = i_{t-1})$$

$$= \sum_{(i_t, \ldots, i_k) \in [m]^{k-t+1}} \|A_S^\dagger\|_F^2 \frac{\overrightarrow{P}(s_1 = i_1, \ldots, s_k = i_k; A)}{\overrightarrow{P}(s_1 = i_1, \ldots, s_{t-1} = i_{t-1}; A)}$$

$$= \frac{\sum_{(i_t, \ldots, i_k) \in [m]^{k-t+1}} \det(A_{\{i_1, \ldots, i_k\}} A_{\{i_1, \ldots, i_k\}}^\top) \|A_{\{i_1, \ldots, i_k\}}^\dagger\|_F^2}{\sum_{(i_t, \ldots, i_k) \in [m]^{k-t+1}} \det(A_{\{i_1, \ldots, i_k\}} A_{\{i_1, \ldots, i_k\}}^\top)}$$

$$= \frac{\sum_{j=1}^n \sum_{(i_t, \ldots, i_k) \in [m]^{k-t+1}} \det(A_{\{i_1, \ldots, i_k\}}^j (A_{\{i_1, \ldots, i_k\}}^j)^\top)}{\sum_{(i_t, \ldots, i_k) \in [m]^{k-t+1}} \det(A_{\{i_1, \ldots, i_k\}} A_{\{i_1, \ldots, i_k\}}^\top)}$$

While the denominator is the (unnormalized) marginal distribution $P(T \subseteq S \mid S \sim P(S; A))$, the numerator is the summation of (unnormalized) marginal distribution $P(T \subseteq S \mid S \sim P(S; A^j))$ for $j = 1, \ldots, n$. By Theorem 2 we can compute this expectation in $\mathcal{O}(nm^3)$ time. $\square$

---

**Algorithm 2** Derandomized Dual Volume Sampling for Column Subset Selection.

---

**Input:** Matrix $A \in \mathbb{R}^{n \times m}$ to sample columns from, $m \leq k \leq n$ the target size
**Output:** Set $S$ such that $|S| = k$ with the guarantee

$$\|A_S^\dagger\|_F^2 \leq \frac{m-n+1}{k-n+1}\|A^\dagger\|_F^2; \quad \|A_S^\dagger\|_2^2 \leq \frac{n(m-n+1)}{k-n+1}\|A^\dagger\|_2^2$$

Initialize $\overrightarrow{S}$ as empty tuple
**for** $i = 1$ to $k$ **do**
    **for** $j \notin \overrightarrow{S}$ **do**
        Compute conditional expectation $E_j = \mathbb{E}\left[\|A_T^\dagger\|_F^2 \mid t_1 = s_1, \ldots, t_{i-1} = s_{i-1}, t_i = j\right]$ with Corollary 4.
    **end for**
    Choose $j = \arg\min_{j \notin \overrightarrow{S}} E_j$
    $\overrightarrow{S} = \overrightarrow{S} \circ j$
**end for**
Output $\overrightarrow{S}$ as a set $S$

---

# E   Greedy Derandomization

**Theorem 14.** *Algorithm 2 is a derandomization of dual volume sampling that selects a set $S$ of columns satisfying*

$$\|A_S^\dagger\|_F^2 \leq \frac{m-n+1}{k-n+1}\|A^\dagger\|_F^2; \quad \|A_S^\dagger\|_2^2 \leq \frac{n(m-n+1)}{k-n+1}\|A^\dagger\|_2^2.$$

*Proof.* Observe that at each iteration $t$, we have

$$\mathbb{E}\left[\|A_T^\dagger\|_F^2 \mid t_1 = s_1, \ldots, t_{i-1} = s_{i-1}\right]$$
$$= \sum_{j \notin \overrightarrow{S}} \overrightarrow{P}(t_i = j \mid t_1 = s_1, \ldots, t_{i-1} = s_{i-1})\mathbb{E}\left[\|A_T^\dagger\|_F^2 \mid t_1 = s_1, \ldots, t_{i-1} = s_{i-1}, t_i = j\right],$$

and we choose $j$ such that $\mathbb{E}\left[\|A_T^\dagger\|_F^2 \mid t_1 = s_1, \ldots, t_{i-1} = s_{i-1}, t_i = j\right]$ is minimized. Since at the beginning we have

$$\mathbb{E}\left[\|A_T^\dagger\|_F^2\right] \leq \frac{m-n+1}{k-n+1}\|A^\dagger\|_F^2; \quad T \sim P(T; A),$$

it follows that the conditional expectation satisfies

$$\mathbb{E}\left[\|A_T^\dagger\|_F^2 \mid t_1 = s_1, \ldots, t_{i-1} = s_{i-1}, t_i = j\right] \leq \frac{m-n+1}{k-n+1}\|A^\dagger\|_F^2.$$

Hence we have

$$\|A_S^\dagger\|_F^2 = \mathbb{E}\left[\|A_T^\dagger\|_F^2 \mid t_1 = s_1, \ldots, t_{k-1} = s_{k-1}, t_k = s_k\right] \leq \frac{m-n+1}{k-n+1}\|A^\dagger\|_F^2.$$

Further, by using standard bounds relating the operator norm to the Frobenius norm, we obtain

$$\|A_S^\dagger\|_2^2 \leq \|A_S^\dagger\|_F^2 \leq \frac{m-n+1}{k-n+1}\|A^\dagger\|_F^2 \leq \frac{n(m-n+1)}{k-n+1}\|A^\dagger\|_2^2.$$

$\square$

# F   Initialization

Set $\varepsilon = \min_{|S|=k} \sigma_n^2(A_S) > 0$, whereby

$$\det(A_S A_S^\top + \varepsilon I_n) = \varepsilon^{n-k} \det(A_S^\top A_S + \varepsilon I_k) \propto VolSmpl\left(S; [A^\top \quad \sqrt{\varepsilon}I_m]^\top\right).$$

The rhs is a distribution induced by volume sampling. Greedily choosing columns of $A$ one by one gives a $k!$ approximation to the maximum volume submatrix [15]. This results in a set $S$ such that

$$
\begin{aligned}
\det(A_S A_S^\top) &\geq \frac{1}{2^n} \det(A_S A_S^\top + \varepsilon I_n) = \frac{1}{2^n \varepsilon^{k-n}} \det(A_S^\top A_S + \varepsilon I_k) \\
&\geq \max_{|S|=k} \frac{1}{2^n k! \varepsilon^{k-n}} \det(A_S^\top A_S + \varepsilon I_k) = \max_{|S|=k} \frac{1}{2^n k!} \det(A_S A_S^\top + \varepsilon I_n) \\
&\geq \frac{1}{2^n k! \binom{m}{k}} \sum_{|S|=k} \det(A_S A_S^\top + \varepsilon I_n) \geq \frac{1}{2^n k! \binom{m}{k}} \sum_{|S|=k} \det(A_S A_S^\top).
\end{aligned}
$$

Thus, $\log P(S; A)^{-1} \geq \log(2^n k! \binom{m}{k}) = \mathcal{O}(k \log m)$. Note that in practice it is hard to set $\varepsilon$ to be exactly $\min_{|S|=k} \sigma_n^2(A_S)$, but a small approximate value suffices.

## G Experiments

We show full results on CompAct(s), CompAct, Abalone and Bank32NH datasets in Figure 2, 3, 4 and 5 respectively. We also run DVS-*, which is $\frac{1}{*}$-generalized DVS algorithm. We observe that decreasing $\beta$ sometimes helps but sometimes not. In Figure 5 we observe that optimization- or greedy-based methods, while taking a huge amount of time to run, perform better than all sampling-based methods, thus for these selection methods, one is not always superior than another.

Figure 2: Results on CompAct(s). Note that `Unif`, `Lev`, `PL` and `DVS` use less than 1 second to finish experiments.

Figure 3: Results on CompAct. Note that `Unif`, `Lev`, `PL` and `DVS` use less than 1 second to finish experiments.

Figure 4: Results on Abalone. Note that `Unif`, `Lev`, `PL` and `DVS` use less than 1 second to finish experiments.

Figure 5: Results on Bank32NH. Note that `Unif`, `Lev`, `PL` and `DVS` use less than 1 second to finish experiments.