[Reviews · NeurIPS 2017]

Reviewer 1



The authors study the problem of sampling subsets of points proportionally to their volume-span. The main contributions are; (a) polynomial time sampling algorithms (using marginals and involved matrix manipulations) and (b) propose a MCMC sampler for volume sampling via a connection between dual volume sampling and the Strongly Rayleigh property of measures. The paper is well written and the main contributions are well explained. The references to prior work is accurate (to the best of my knowledge). In my opinion, the connection between dual volume sampling and Strongly Rayleigh measures is interesting. I have a few questions that I would like to share with the authors. Major comments: 1) In L158-162, you state that determinantal point processes are known to be Strongly Rayleigh measures. If so, is Theorem 6 well known or it is part of your contributions here? Please clarify it. 2) Algorithm 1, "while not mixed do". Is it possible to have a stopping criteria here? Can you decide when to stop? If not, I would suggest to have an input parameter "number of iterations" 3) Experiments: Legends on first 2 plots are missing 4) Experiments: Please replot the "time-error trade-off" plot. Use different scale. Minor comments: 1) L110: Spell out "Thm. 2." to Theorem

Reviewer 2



Summary: The authors present an algorithm for Dual Volume Sampling (DVS). This is an approach proposed by Avron and Boutsidis for experimental design and picks a subset S of k columns of a fat matrix A with probability proportional to det(A_S A_S') where A_S is the matrix restricted to columns in S. Avron and Boutsidis also showed that such a sampling provides a good approximation for E and A optimal experimental designs. Their (first) algorithm samples columns iteratively. The authors derive explicit yet efficiently computable formulae for the marginal probabilities at each step. Next, the authors prove that the DVS distribution satisfies a powerful property known as the "Strongly Rayleigh" property, which almost immediately implies a fast mixing Markov chain for DVS sampling. Finally, the authors present experimental results for experiment design in regression problems. These algorithms demonstrate a clear advantage in terms of both running time and the error achieved. Opinion: I think the authors make a fundamental contribution to understanding DVS and the first algorithms for efficiently sampling from them. The theoretical bounds on the algorithm are not very impressive, but the experiments demonstrate that for a good range of parameters, they offer a strong time advantage over several other methods used in practice. Overall, I recommend acceptance. Small typos: 181: "an diagonal" typo 183: S <= T should be T <= S Edit : I have read the authors' feedback, and the review stands unchanged.

Reviewer 3



The paper studies efficient algorithms for sampling from a determinantal distribution that the authors call "dual volume sampling", for selecting a subset of columns from a matrix. The main results in the paper are two sampling algorithms for selecting k columns from an n x m matrix (n<=k<=m): - an exact sampling procedure with time complexity O(k m^4) - an approximate sampling, with time complexity O(k^3 n^2 m) (ignoring log terms). The approximate sampling algorithm makes use of Strongly Rayleigh measures, a technique previously used for approximately sampling from closely related determinantal distributions, like Determinantal Point Processes. Compared to the exact sampling, it offers a better dependence on m, at the cost of the accuracy of sampling, and a worse dependence on k and n. The authors perform some preliminary experiments comparing their method to other subset selection techniques for linear regression. It is worth noting that the plots are not very informative in comparing dual volume sampling to leverage score sampling which is to my mind the most relevant baseline. This determinantal distribution was previously discussed and motivated in [1], who referred to it just as "volume sampling", and suggested potential applications in linear regression, experimental design, clustering, etc., however without providing a polynomial time algorithm, except for the case of k=n. The paper addresses an important topic that is relevant to the NIPS community. The results are interesting, however the contribution is somewhat limited. To my knowledge, the algorithms (in particular, the exact sampling algorithm) are much slower compared to Leverage Score sampling, and the authors did not make a compelling argument for why DVS is better suited for the task of experimental design. [1] Avron and Boutsidis. Faster Subset Selection For Matrices and Applications, 2013. https://arxiv.org/abs/1201.0127